# Nearby armed conflict affects girls' education in Africa

**Xiao Hui Tai** *

Department of Statistics, University of California Davis, Davis, California, United States of America

* xtai@ucdavis.edu

## Abstract

Female education is a crucial input to women's agency and empowerment, and has wide-ranging impacts, from improved labor market outcomes to reducing child mortality. Existing gender-specific evidence on the effect of armed conflict on education is conflict-specific and mixed. We link granular data on conflict events to georeferenced survey data on educational attainment from 28 countries in Africa, and use a regression-based approach to estimate the local effect of conflict exposure on female years of schooling. We find that conflict events occurring within 25 kilometers during a female child's primary school years reduces years of schooling by 0.4 years by adolescence. We do not find the same effect for males. Exposure to only low intensity conflict events with at most two casualties has persistent negative and significant effects. Consecutive years of conflict, however, can have positive effects in later years, which offset earlier negative effects, suggesting a habituation to violence. In the past two decades, we estimate excess child mortality in Africa associated with the indirect channel of women's education to be similar in magnitude to the number of direct child casualties due to conflict.

**Data Availability Statement:** Third party data was obtained for this study from the Uppsala Conflict Data Program (https://ucdp.uu.se/downloads/) and the DHS Program. Data may be requested from the DHS Program after creating an account and submitting a concept note. More access

## Introduction

Female education, the focus of the United Nations Sustainable Development Goals 4 and 5, is an important goal and marker of development. Educating women in developing countries has been described as having a "catalytic effect on every dimension of development" [1], producing economic [2] and social benefits, such as reducing fertility [3] and child mortality [4].

Over four-fifths of countries in Africa, representing 16% of the world's population, have experienced armed conflict in recent decades [5]. The indirect and longer-term costs of such conflict, in particular to women and children, are substantial [6, 7]. Despite the large body of literature on the detrimental effects of armed conflict on education [8], gender-specific effects have largely been studied with respect to specific conflicts and have yielded contradictory results [9, 10]. Developing a broader base of evidence on the effect of conflict on female education is crucial to informing the design of policies to mitigate its impacts, particularly in countries lacking previous studies or where the nature of conflict differs from previous work.

How gender affects the relationship between conflict and education is theoretically ambiguous. On the one hand, boys may be more severely impacted, since they previously enjoyed an

information can be found on the DHS Program website (https://dhsprogram.com/data/Access-Instructions.cfm). The authors confirm that interested researchers would be able to access these data in the same manner as the authors. The authors also confirm that they had no special access privileges that others would not have.

**Funding:** The author(s) received no specific funding for this work.

**Competing interests:** The authors have declared that no competing interests exist.

advantage in terms of education (e.g., in Rwanda [11] and Burundi [10]), or are sent to battle (e.g., in Uganda [12]). On the other hand, girls may be more severely impacted, if families concentrate limited resources on boys (e.g., in Guatemala [13]) or keep girls home due to safety reasons (e.g., in Tajikstan [9]).

Here, we combine several publicly available sources of georeferenced data for the African continent to estimate the effect of nearby violent events on female education, measured in terms of years of schooling. One of our main contributions is to produce evidence with a broad scope in space and time, as well as a high spatial resolution, allowing us to isolate localized effects. Existing work either focuses on specific conflicts or adopts a macro, country-level approach [14], which misses the key distinction between conflict-affected countries and conflict-affected areas. This distinction is particularly important in the context of schooling, which requires daily attendance and is likely to be sensitive to local effects, and conflict in Africa, which has been highly localized (Fig 1). Our work adds to recent literature that estimates effects of nearby armed conflict in Africa, but that focuses on public health outcomes such as child mortality [7, 15–19].

We use survey data from the Demographic and Health Surveys (DHS), which provides internationally comparable population and health data in most developing countries, including 42 in Africa. In 38 of these countries, at least one available survey contains geo-coordinates of households surveyed, enabling us to link these to georeferenced data on conflict events and measure conflict exposure at a local level. We focus on exposure to conflict during the primary school ages, since years of schooling in Africa is low (Fig 1) and primary school completion is a critical educational milestone [21]. Our main measure of conflict exposure is whether or not an individual experiences a violent event within 25 km—a reasonable choice as it is roughly the distance a person can walk in one day [18] and is further than distances to schools for majority of school-age children in Africa. More discussion about this threshold is in *Materials and methods*. We measure educational outcomes at late adolescence, when primary education is mostly complete [22]; this captures age-specific disruptions to human capital formation that occur during childhood [8]. Our measurement strategy isolates the effect of local conflict exposure on *non-migrants*, a group we are able to accurately measure conflict exposure for (full details in Materials and methods). This means we capture effects *not* due to migration and displacement—known consequences of conflict (e.g., [23, 24]) that could have both positive and negative impacts on education [10, 25]—an issue we discuss further in *Discussion*.

Since many factors may contribute to associations between conflict exposure and years of schooling, we use a regression framework that accounts for location-specific factors (e.g., certain areas being less likely to experience conflict and having a higher female educational attainment), cohort effects (e.g., global events), time-varying factors such as climate and rainfall, and household characteristics such as wealth. We thus compare children in the same location that experienced conflict during their primary school years, to children who did not—a similar strategy to other recent work that examines the effect of conflict on child mortality [15], maternal mortality [16] and immunizations [18]. To interpret the estimates as causal, we rely on the assumption that there are no remaining unmeasured confounders that affect both conflict and educational outcomes, that vary within the same location over time (birth year). We believe this assumption to be plausible but acknowledge the limitations of inferring causal effects from observational data (see Materials and methods).

We find that female children who were exposed to at least one conflict event during their primary school years lost 0.38 years of schooling by adolescence. The effect for males was positive but insignificant. For girls, we document effects for conflict exposure up to 50 km away, with the largest impacts at the ages of 6 and 11. Even small exposures to conflict—violent events with at most two total casualties in seven years—results in fewer overall years of

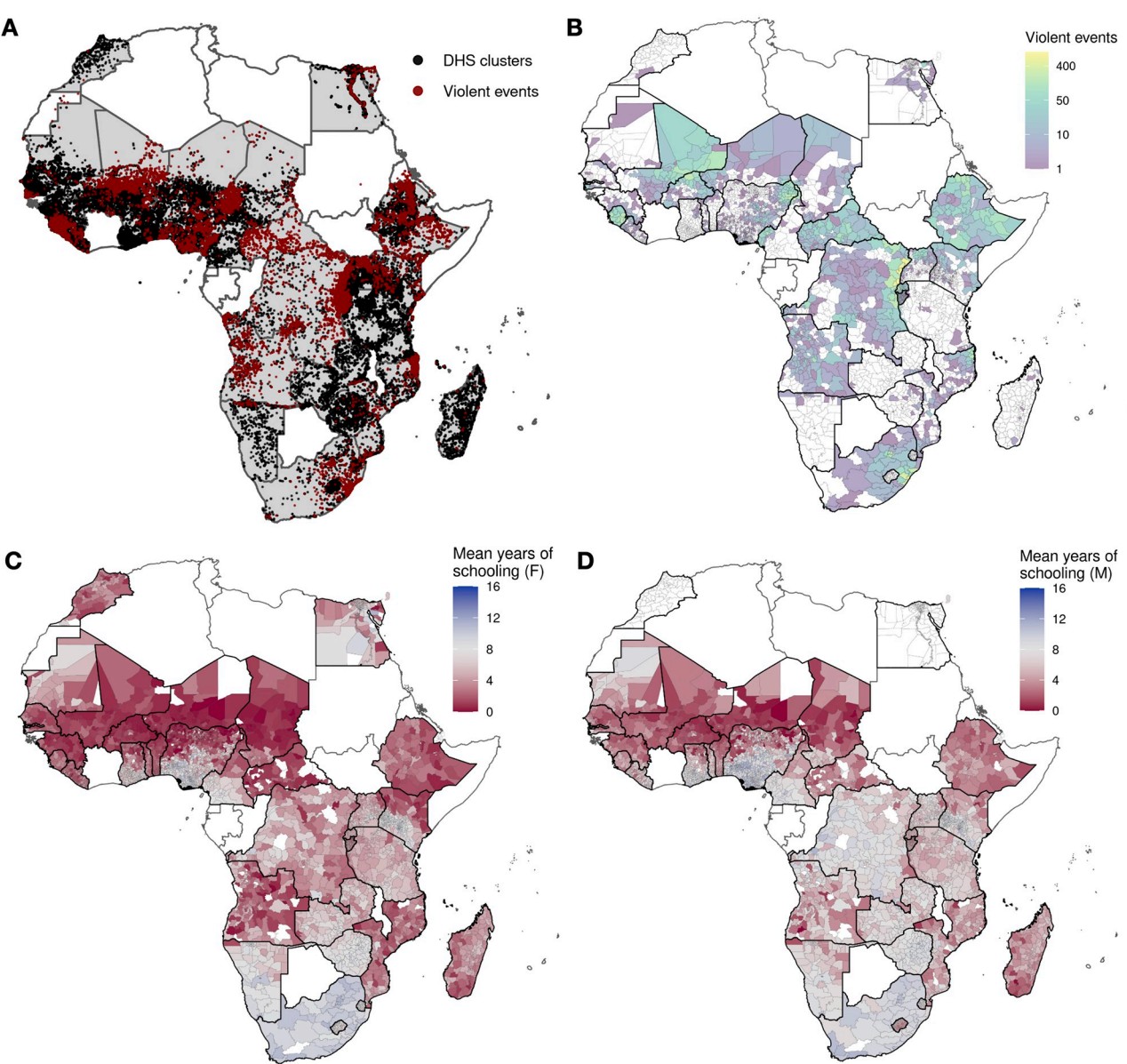

**Fig 1. Geographical distribution of survey data and conflict events. A**. There are 35 countries in Africa with both DHS data and conflict data (shaded in gray). In these countries, the locations of 56,623 DHS clusters (black dots) from 118 surveys (conducted in 1986–2022), and 31,973 violent events from 1989–2022 (red dots) are shown. **B**. Counts of violent events in A for each administrative level 2 region. **C**. Mean years of schooling for female survey respondents (aged 15–49) for each administrative level 2 region. **D**. Same as C, but for males. Regions with no data are colored in white. Administrative boundaries are from [20].

schooling. On the other hand, recurring violence, defined as events occurring in consecutive years, have positive effects in later years of exposure (that offset negative effects in earlier years), suggesting a habituation to conflict. Locations that are wealthy but have lower baseline levels of female educational attainment see the most negative effects. There are also regional differences in sub-Saharan Africa (SSA), with Eastern SSA experiencing less severe effects.

Prior work has suggested some mechanisms by which conflict affects girls' schooling: school closures, safety, reallocation of household resources, child marriages, and death of

parents [9, 13, 26–28]. The DHS data are quite rich and allow us to test several of these—we do not find that conflict exposure led to increased child marriages or death of parents. Rather, the results relating to habituation and location-specific heterogeneity suggest that concerns about safety or a reallocation of household resources are likely to be driving the results that we observe. Additional data and more research is required, however, to confirm these hypotheses.

Finally, we highlight one long-term implication of a decrease in educational attainment among females. The negative relationship between mothers' education and child mortality is well-established and has been described as "one of the most consistent and powerful findings in public health" [4]—using estimates of the magnitude of this relationship, we estimate that the reduction in female years of schooling due to armed conflict is associated with 6,800 to 10,500 excess under-5 deaths per year from 2000 to 2019 in Africa. In almost all of these years, this exceeds the number of direct deaths due to conflict and terrorism, which according to the 2019 Global Burden of Disease Survey [29], range from 1,500 to 15,100. This underscores the importance of quantifying not only the more salient direct costs, but also the indirect and longer-term costs of conflict.

## Results

### Conflict exposure and educational attainment

We combine household survey data from the DHS with information on fatal conflict events collected by the Uppsala Conflict Data Program (UCDP). This a publicly available database of fatal violent events that are perpetrated by organized actors, containing information about the location, timing and estimated number of deaths [30]. There are 35 African countries with geo-coded household survey data that have recorded at least one conflict event from 1989 to 2022. DHS data are geo-coded by "cluster," roughly a census enumeration area. In total, these data consist of 31,973 violent events (after filtering out events with insufficient geographical precision; see Materials and methods) and 118 DHS surveys from 1986 to 2022 with interviews of 1,938,424 individuals (men and women) from 56,623 clusters. The spatial distribution of conflict events and DHS clusters, that form the basis of our sample construction, are shown in Fig 1A.

In the recent three decades, conflict has been widespread, affecting four-fifths of countries and 30% of administrative level 2 regions (1,731 out of 5,818) in those countries (Fig 1B). Violence is of low intensity in majority of these regions, with over half (953 regions) experiencing a total of four or fewer events during the entire time period of 34 years.

The DHS data contain responses of women aged 15–49, and in a subset of the surveys, men of similar ages, to questions including age, educational attainment, and years living in residence. The mean years of schooling is 5.3 years for female survey respondents, and 6.6 years for male respondents. 60% of administrative level 2 regions have mean years of schooling of 6 years or less for females, and 44% for males (Fig 1C and 1D).

### Effect of conflict on female education

To quantify the relationship between conflict events (Fig 1B) and years of schooling (Fig 1C and 1D), we combine the UCDP and DHS data spatially and temporally. We measure conflict exposure by whether or not a violent event occurs within 25 km during a child's primary school years, age 6 to 12. We measure years of schooling at age 15–18, for individuals who had lived in their current residence at least since age 6—a group that we are able to accurately estimate the effect of local conflict exposure on. A full discussion of our measurement strategy is in *Materials and methods*, and a full list of DHS countries and years included in the analysis is in S1 Table.

Using a regression framework to control for factors that affect both conflict occurrence and education, we find that on average, females that were exposed to conflict experienced a significant reduction of 0.38 (95% C.I.: (-0.57, -0.18), $P < 0.001$) years of schooling by ages 15–18 (Fig 2A). For males, there is no significant effect (point estimate of 0.17, 95% C.I.: (-0.29, 0.63), $P = 0.47$). The difference between the effect for females and males is statistically significantly different from zero (point estimate of -0.54, 95% C.I.: (-1.04, -0.046), $P = 0.032$). The remaining analysis focuses on the effect for females.

Conflict events occurring within up to 50 km of the cluster's GPS coordinates have a significant negative impact on girls' schooling (Fig 2B). This is more localized than the effect of conflict exposure on public health outcomes, a finding we discuss in *Discussion*. Effects for 0–25 km are the most precisely estimated; the remaining analysis uses this distance threshold.

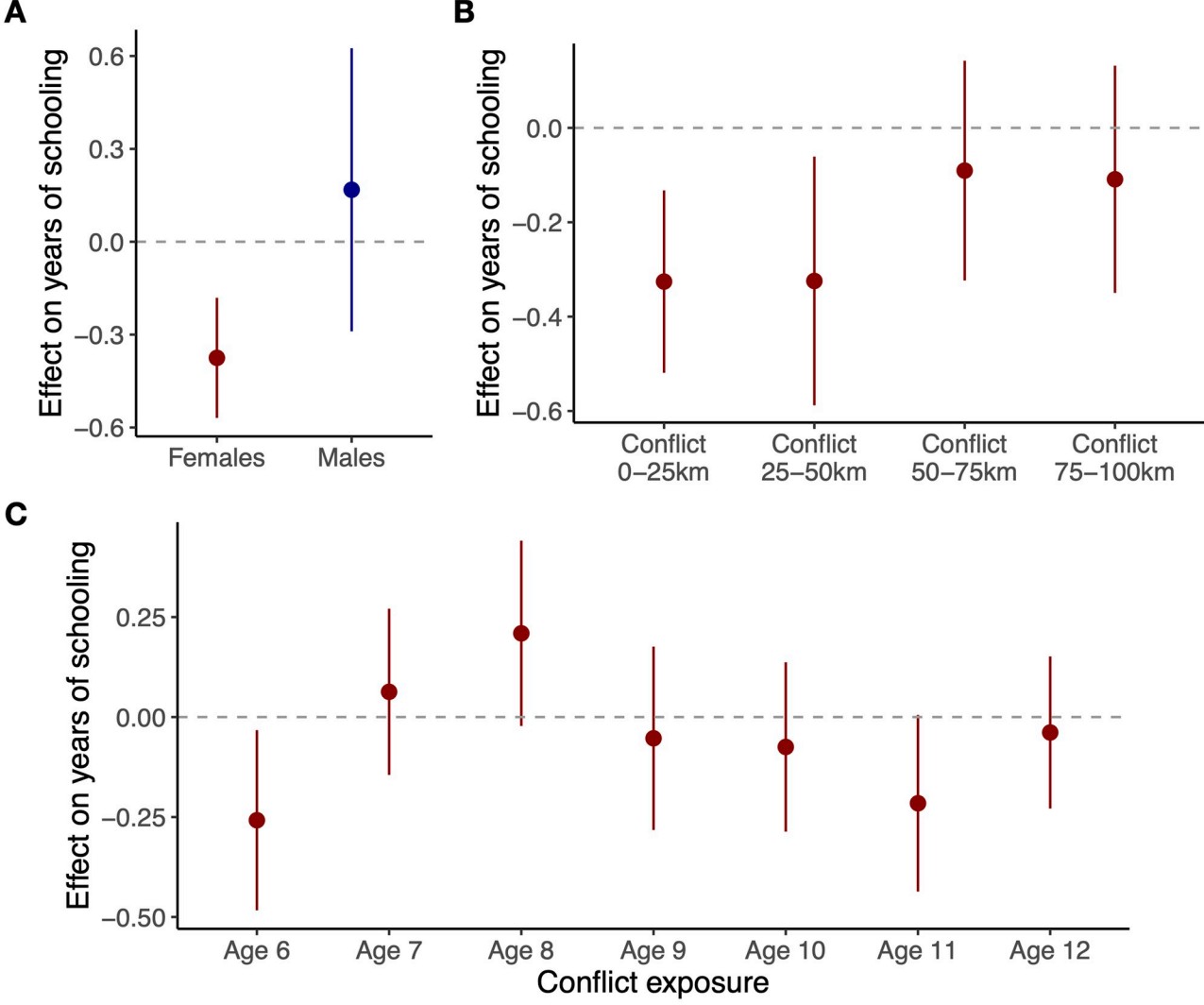

**Fig 2. Effect of armed conflict exposure during the ages of 6 to 12 on years of schooling at age 15–18. A**: The estimated effect for females and males. Regression coefficients control for location-specific and cohort characteristics, time-varying factors and household characteristics. **B**: For females, effect of conflict at various distance thresholds on years of schooling. **C**: Effect of conflict exposure at individual ages for females. Conflict in each year is defined as the occurrence of a violent event within 25 km in each year that the respondent turned 6 to 12. Bars are 95% confidence intervals.

Examining the effect of conflict at each primary school age from 6 to 12, we find a significant negative effect at age 6 (-0.26 years, 95% C.I.: (-0.48, -0.033), $P = 0.025$), and a smaller effect at age 11 (-0.22 years, 95% C.I.: (-0.44, 0.0052), $P = 0.056$)) (Fig 2C). These correspond to the "milestone" years of primary school enrollment and completion, suggesting that girls' education is most susceptible to being disrupted at these ages. Consistent with these results, we find that conflict at age 6 increases the odds of having no education by a multiplicative factor of 1.87 (95% C.I.: (1.05, 3.35), $P = 0.034$), and conflict at age 11 increases the odds of having an incomplete primary education by a multiplicative factor of 1.44 (95% C.I.: (0.96, 2.17), $P = 0.081$) (S1 Fig). These findings are intuitive and complement evidence in the literature that conflict exposure reduces school enrollment and completion [8], and that conflict onset during the initial primary schools are crucial [10].

In analysis described in *Materials and methods* and presented in the supplementary figures and tables, we conduct a variety of robustness checks to ensure that these results are not unduly influenced by how we construct our sample (e.g., the residence condition), modeling assumptions (e.g., different covariates and distance thresholds), as well as our choice of outcome variable (S2–S4 Tables). Our main results do not change under these alternate specifications.

## Acclimation to conflict

Consistent with the low average frequency of violence observed in Fig 1B, we observe that the median person in the sample who was exposed to violence experienced a median of three events and 15 total casualties (Fig 3A). In the seven-year period in which they were 6–12, 49% experienced conflict only in a single year (Fig 3B). At the other extreme, 10% of those exposed experienced 33 or more events and 476 or more casualties.

How might these exposures with different chronicity and intensity change the effect on years of schooling? We first consider the effect of only the lowest casualty violence: up to two casualties in the seven-year period, the bottom quintile of the number of casualties experienced. 22% of individuals who were exposed to violence experienced violence of this type, and perhaps surprisingly, we find a significantly negative effect of a reduction of 0.33 years of schooling (95% C.I.: (-0.65, -0.0079), $P = 0.045$) for this group (Fig 3C). This is similar in magnitude to the effect for those that experienced higher-intensity conflict (point estimate -0.40, 95% C.I.: (-0.64, -0.16), $P = 0.0013$); the difference is not statistically significant (95% C.I. for difference: (-0.34, 0.48), $P = 0.74$). Further, extending the analysis in Fig 2C by disaggregating conflict exposure into new conflict (violence did not occur in the previous year) and recurring conflict (violence occurred in the previous year), we observe that new conflict has the most negative effect at age 6 (Fig 3D). At age 8, recurring conflict has a positive effect (point estimate 0.35, 95% C.I.: (0.052, 0.64), $P = 0.021$)—perhaps suggesting deferred school enrollment during conflict. Both these results are suggestive of some degree of habituation to conflict. Note that the positive coefficient in Fig 3D does not more generally suggest that conflict increases final educational attainment—the definition of recurring conflict is that it has also occurred in the previous year, and the positive coefficient can be interpreted as offsetting negative effects in previous years. More generally, estimating the overall effect on educational attainment will require considering a linear combination of coefficient estimates from ages 6 to 12.

## Location-specific heterogeneity in effects

Next, we investigate whether the negative overall effect on female years of schooling changes depending on location characteristics. We consider whether or not a location is urban or rural, relative wealth, level of educational attainment in the older generation, gender gap in

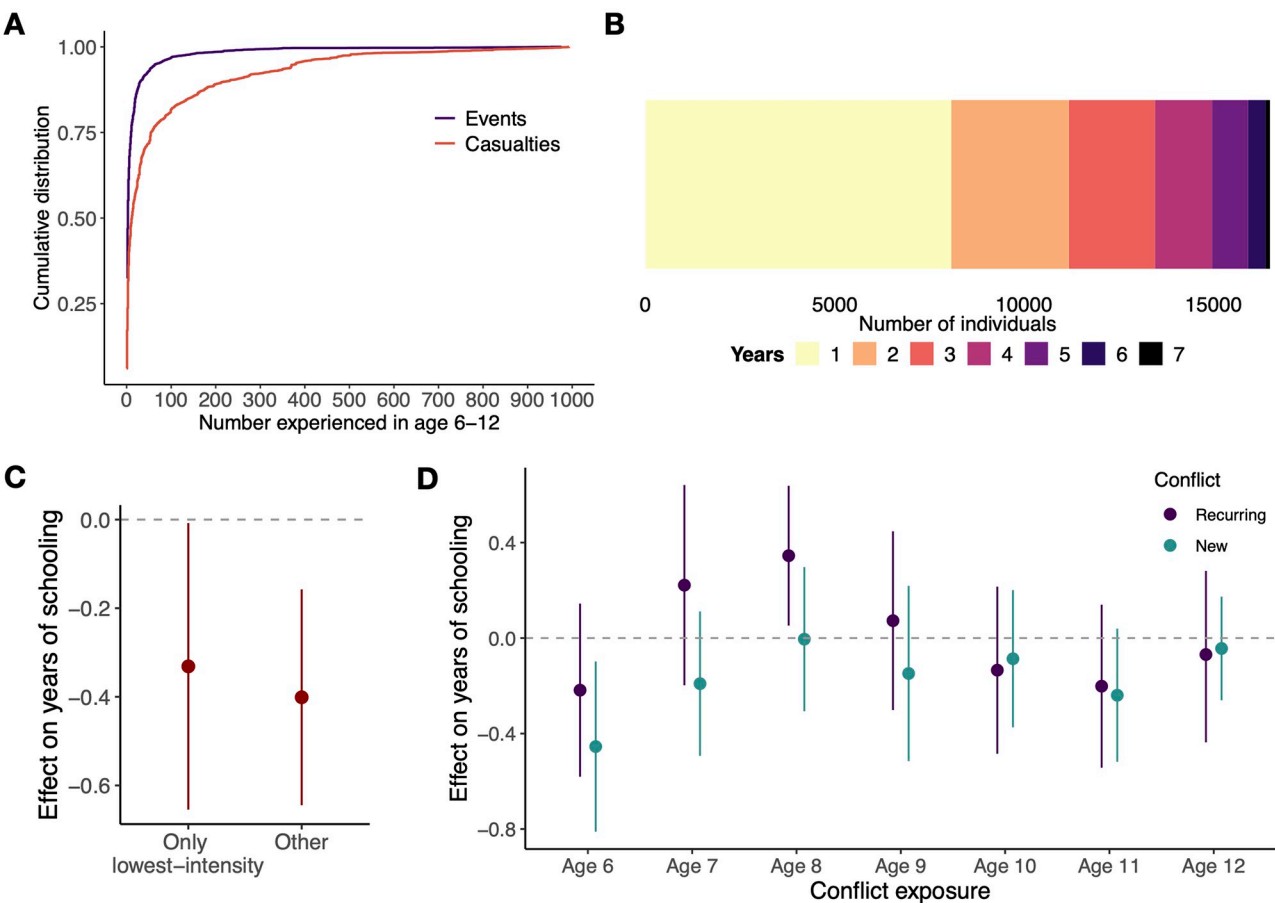

**Fig 3. Different impacts for conflict exposures of different intensity and chronicity. A**: Cumulative distribution function of the number of violent events and casualties experienced, for girls who experienced conflict within 25 km during the ages of 6–12. **B**: Total number of years between the ages of 6–12 experiencing violent events, for the same individuals in A. **C**: Effect of exposure to the lowest intensity of violence (up to 2 casualties, the bottom quintile of the number of casualties experienced), versus other exposures (3 or more casualties), on years of schooling. **D**: Effect of conflict exposure at individual ages, for recurring conflict (experienced violence in the previous year) versus new conflict (no events in the previous year). Bars are 95% confidence intervals.

educational attainment, and geographical region. Some of these are commonly-studied factors modifying the effects of armed conflict; for example for public health outcomes, urban areas, wealthier areas and more highly educated areas generally experience smaller impacts [15, 17, 19]. For wealth and education, we use the relative rankings within our sample of the wealth of surveyed households, and education level of older females. For gender gap, we compare the mean years of schooling among males and females. Geographical regions are as defined in the Global Burden of Disease (GBD) Study, designed to be geographically close and similar in socio-demographics and epidemiology [29]. Full details are in *Materials and methods*.

We find that whether an area is rural or urban does not change the effect of conflict on education (Fig 4). Wealthier clusters are more severely negatively impacted, as are regions with the lowest baseline female education levels. The result on wealth is surprising for two reasons. First, we might expect wealth to be protective, in which case the opposite relationship should hold. However, it is possible that children in the poorest locations, in particular girls, are less likely to attend school regardless of the occurrence of conflict, so the effect of conflict on girls is small [10]. Second, wealth and education tend to be correlated, so smaller negative effects

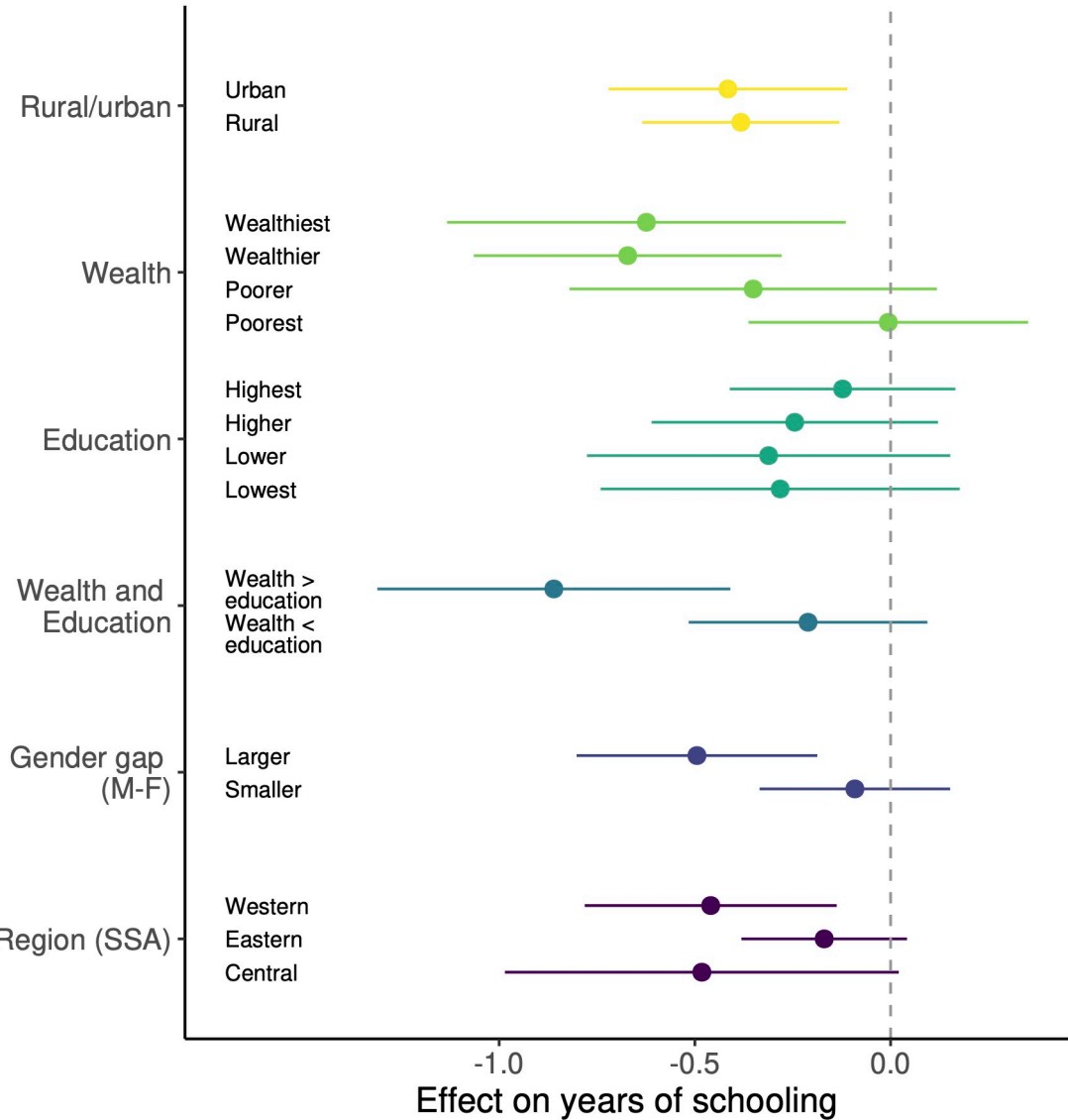

**Fig 4. How the effect of conflict exposure depends on location characteristics.** Coefficient estimates represent the effect of conflict exposure on female years of schooling, estimated separately for each type of location: rural vs. urban (yellow bars), wealth (light green bars), education (dark green bars), gender gap in education (violet bars) and region in sub-Saharan Africa (purple bars). For wealth and education, locations are split into four quartiles by relative household asset-based wealth, and mean years of schooling for older females. Teal bars represent locations where the wealth quartile exceeds the education quartile, and vice versa. For gender gap, locations are split into whether they have gaps above ('larger') or below the median ('smaller'). Regions are as defined by the GBD study. Bars are 95% confidence intervals.

with more education should predict smaller negative effects with greater wealth. Examining the number of clusters in each wealth and education quartile, the wealthiest clusters do indeed have the highest baseline educational levels, and the poorest clusters have the lowest educational levels (S2 Fig). These findings suggest an interaction between wealth and education.

To investigate this relationship further, we consider locations where the wealth quartile exceeds the educational attainment quartile, and where the opposite is true. The former might be thought of as locations in which female education lags behind wealth. We find that the most negative impacts indeed occur among these locations (first teal bar in Fig 4; point estimate

-0.86, 95% C.I.: (-1.31, -0.41), *P* < 0.001). The pattern of these results is robust to sensitivity checks that address the concern that the wealth of a cluster is endogenous and affected by conflict (S3 Fig).

Additionally, more severe negative impacts are estimated in locations where the male-female gender gap is larger (violet bars in Fig 4). In locations where the gap is above the median of 1.2 years, conflict exposure reduces years of schooling by 0.49 years (95% C.I.: (-0.80, -0.19), *P* = 0.0016), compared to a reduction of 0.091 years (95% C.I.: (-0.33, 0.15), *P* = 0.46) for locations where the gap is below the median. This means that in areas where females have a smaller existing educational disadvantage, they are also less negatively affected by conflict, whereas in areas where they face a larger disadvantage, females are more severely impacted by conflict.

Finally, the estimated effect in Eastern SSA (-0.17; 95% CI: (-0.38, 0.042), *P* = 0.12) is much smaller in magnitude than in Western SSA (-0.46; 95% CI: (-0.78, -0.14), *P* = 0.0052) and Central SSA (-0.48; 95% CI: (-0.99, 0.021), *P* = 0.061) (purple bars in Fig 4). While these differences are not statistically significant as they are not sufficiently precisely estimated, the findings are substantively interesting. These regional differences may partially be explained by gender gaps in education—locations in Eastern SSA have smaller gender gaps in education compared to those in Central and Western SSA (S4 Fig). In the regression sample, 88.5% and 61.1% of individuals in Central and Western SSA are classified as having male-female gaps above the median, while this number is 23.3% in Eastern SSA.

## Mechanisms of the impact of armed conflict

To summarize the preceding results, we find that female education is negatively affected by conflict exposure, with larger impacts at closer distances and at specific ages. Effects are observed even for infrequent and low-intensity violence; on the other hand, there is evidence of habituation to violence. Effects are most severe in locations with high relative wealth but where women have lower levels of education, and in locations with larger gender gaps in educational attainment. What do these findings point to in terms of mechanisms in which conflict affects female schooling?

Potential mechanisms proposed in the literature can be grouped into supply-side and demand-side factors. For the non-displaced population, which is the focus of our analysis, supply-side factors include destruction of infrastructure and school closures [26]. Demand-side factors include concerns about safety [9], reallocation of household resources to boys [13], child marriages [27], and death of parents [28] (with girls more negatively impacted than boys). The evidence we have presented favors demand-side factors, since supply-side factors are more likely to have a uniform impact on all school-going children, not only girls at particular ages. More intense fighting is likely to worsen supply-side conditions, resulting in more severe impacts; there is limited evidence to support this claim.

Among the possible demand-side factors, in additional analyses (S5 and S6 Tables) we do not find evidence of increased child marriages or death of parents among girls exposed to conflict during the ages of 6–12. Given our focus on adolescents who had not moved since age 6, child marriages are unlikely to be the primary mechanism. For the same reason, and coupled with the negative effect observed for low-casualty violence, neither is the death of parents. The channels that are likely to be driving our results are concerns about safety or a reallocation of household resources. It is plausible that households in wealthy areas with low female educational attainment, or in areas with large gender gaps in terms of education, perceive female education to be of lower importance, and keep girls at home at the first sign of conflict.

While our data do not allow us to fully test all potential mechanisms, these are important directions for future work. One potential avenue is to explore the possibility of using recently

compiled, granular school-level data [31] and methods for monitoring war-time building destruction [32], to test hypotheses about destruction of infrastructure and school closures. Next, while the DHS data do not capture motivations, concerns about safety and household allocation of resources might be explored by combining DHS data with other survey data, such as from the Afrobarometer (https://www.afrobarometer.org/). Afrobarometer has conducted surveys in 40 African countries on economic, political and social topics, including safety and well-being; a subset of data are geolocated and have been successfully linked in other studies to DHS data (e.g., [33]), making this a promising avenue.

### Implications on child mortality

The overall impact we find is that conflict exposure during primary school reduces female years of schooling in adolescence by 0.38 years. What downstream implications might this have? Given the prominence of child survival in the international development agenda [34], and the documented strength in the negative relationship between mother's education and child mortality [4], we highlight the impact of a loss in educational attainment due to conflict, on child mortality.

We conduct the following back-of-the-envelope calculation for mothers in Africa in the recent two decades. Since some of these mothers would have experienced conflict during their childhood, they would have lost some years of schooling, compared to the conflict-free counterfactual. Children of these mothers would have had higher observed mortality rates, if established negative relationships between mother's education and child mortality hold. This translates into a larger total number of under-5 deaths, an excess over what would have occurred under the conflict-free alternative. We apply estimates from our analysis of the effect of conflict on female education, and of the magnitude of the effect of mother's education on child mortality from a recent global systematic meta-analysis [35], to yearly under-5 mortality rates in Africa from the 2019 GBD study. This calculation requires several assumptions that we explain in detail in *Materials and methods*, along with associated limitations. If, for instance, losses in schooling are reversed between adolescence and motherhood, we would overestimate the number of excess deaths, although we do not expect this to be a major concern (S7 Table).

We find that in the recent two decades, the effect on child mortality ranges from 6,800 to 10,500 excess under-5 deaths per year (Fig 5). Low estimates, which take into account uncertainties in the estimates (Materials and methods) range from 2,500 to 4,500. These low estimates are on the same order of magnitude as the number of direct under-5 casualties due to conflict and terrorism according to the GBD study, and exceed the number of direct casualties in 9 of the 20 years. A caveat in this analysis is that the relationship between mother's education and child mortality, while strong, does not necessarily capture a causal relationship, and is not guaranteed to hold in this context. For example, lower mother's education due to conflict might be mediated in other ways that reduce its impact on child deaths, if for instance, local post-conflict interventions aid recovery and prevent child deaths.

### Discussion

Our work demonstrates an approach to studying the effect of conflict on schooling that is both fine-grained and broad in geographical scope. While previous work has focused on single countries, finding that the gender-specific impacts of conflict on educational outcomes are highly context-specific, we find that in recent decades, there is a robust negative effect on girls' schooling in Africa. Boys' schooling is not similarly affected.

The magnitude of effect that we estimate is a reduction of roughly 0.4 years of schooling for girls. This is largely consistent with other work that studies individual contexts. A review

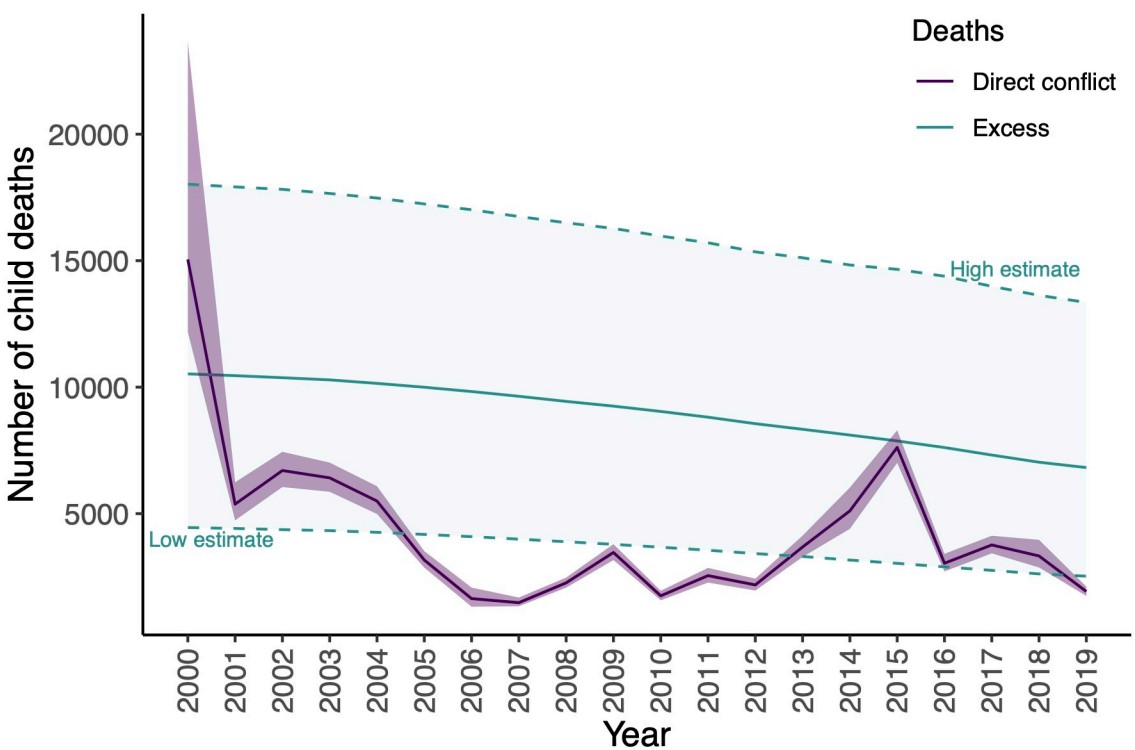

**Fig 5. Direct child deaths due to conflict and estimated excess indirect deaths due to losses in mothers' education in Africa.** In purple are the number of direct child (under-5) casualties due to conflict and terrorism, with bands as low and high estimates, according to the 2019 Global Burden of Disease (GBD) Study. In green are the number of excess deaths due to the indirect channel of conflict reducing female education, which in turn increases child mortality, compared to the conflict-free counterfactual. Green dashed lines represent low and high estimates, taking into account uncertainties in estimates.

article by [8] cites magnitudes for overall (non-gender specific) effects on years of schooling from 0.2 up to 3.5 years over more than a dozen contexts, with most estimates being under 1 year. Gender-specific effects are of a similar magnitude (e.g., 0.5 and 0.3 fewer years of schooling for school-age boys and girls in Rwanda [11]). While one might expect effects to be attenuated when studying larger geographical extents, a granular 25 km conflict exposure measure ensures that our estimates remain localized and not averaged over larger units that include both areas with and without conflict.

Another advantage of granular data is that it allows us to examine the effect of distance to conflict, which is not possible with existing work that considers administrative units [10, 36, 37], households or communities [9]. We find that effects of conflict exposure are felt up to 50 km from the location of violence; this effect is more localized than that for public health outcomes, which have effects up to 100–150 km [15, 19]. This is perhaps unsurprising given that schools, requiring daily attendance, are expected to be closer to homes, whereas people might seek care in medical facilities further from their place of residence, generating larger spatial spillovers for conflict. Illustrative statistics support this hypothesis: World Bank data find that 73% of school-age children had schools within 2 km from their homes [26], while a meta-analysis finds mean distances of 15 km to specified facilities likely with skilled delivery care [38].

We find evidence of negative effects even for conflict exposures with small number of casualties. Related literature suggests that perceived threats, even through indirect exposures to

conflict (vicarious experiences of family members or close friends), harm psychological well-being [39]. This explanation is consistent with our hypothesis that conflict exposure triggers concerns about safety that result in girls not enrolling in or attending school. Despite these negative impacts for low-intensity violence, we find evidence of acclimation to conflict when it recurs. People have been documented to acclimate to conflict in diverse contexts such as during the second Palestinian intifada, the recent war in Ukraine, and the Mexican drug war [40–42]. Behavioral changes are observed, such as ignoring sounds of gunfire and reducing responses to public warning systems. Here the negative impacts for low-intensity violence, and positive impacts for subsequent recurring violence suggests adaptive behaviors to violence. A sudden threat to safety is likely to be disruptive to female schooling, while violence that results in delayed school enrollment may be reversed if violence recurs.

Perhaps surprisingly, our results on location-specific heterogeneity in effects find no difference between rural and urban areas, and that wealth alone is not protective of girls' education. Rural and poor areas are thought to be more vulnerable, with poorer educational infrastructure and more children out of school [26]. The absence of differences in effects may be explained by low levels of school attendance for girls in these locations, regardless of conflict occurrence. Interestingly, the most severe effects are observed in more wealthy clusters with lower educational levels of older females, suggesting that baseline female educational attainment is more important than wealth in moderating the effect of armed conflict on girls' schooling. Although wealth and education are often regarded as interchangeable indicators of socioeconomic status, education of mothers or parents have been found to be more important than wealth in influencing children's academic achievement [43], child intelligence [44] and reducing child mortality [45]. Our findings dovetail with these studies in education, child psychology and demography. Additionally, the larger negative effects estimated in areas where the male-female gender gap is larger lend support to the hypothesis that in areas where female education is perceived to be of a higher importance, girls are more insulated from the impacts of conflict on schooling.

In summary, these findings add important nuances to our understanding of the geographical extent, timing, and location-specific heterogeneity of the effect of conflict exposure on female education. These provide information that can be used to target interventions at individuals that may be most vulnerable. For example, an intervention could be designed with the goal of ensuring that girls at age 6 enroll in primary school at equal rates as boys, despite the occurrence of conflict. Further work should be done to understand the mechanisms at a more detailed and local level, which will inform the design and evaluation of specific interventions. This will enhance our ability to make evidence-based policy recommendations.

Female education has important implications on societies and on development. It follows that a decrease in educational attainment due to conflict exposure might have wide-ranging downstream impacts. We focus on the effects on child mortality, contributing to the literature on estimating the indirect deaths due to conflict. The ratio of indirect to direct conflict deaths has often been quoted as 9:1, but this statistic in fact has little empirical basis [46]. Here, isolating the impact of conflict exposure on child mortality through the mothers' education channel, we find that indirect child deaths are roughly on the same magnitude as direct child deaths. More generally, indirect effects are often underestimated [47] and interact with each other in ways that are not well-understood [48]; we contribute evidence that brings us closer towards painting a full picture of the indirect costs of conflict.

This study has several limitations. First, our main analysis focuses on non-migrants. Migration and forced displacement are well-known consequences of armed conflict, and are important mechanisms through which conflict affects education. The direction of the impact of displacement on education is theoretically ambiguous. On one hand, migrants are vulnerable

populations whose education is likely to be disrupted. On the other hand, uprootedness has been found to lead to investment in human capital, as preferences are shifted away from material possessions [25]. In addition, migration may introduce selection effects, with those that moved being different from those who did not. DHS data do not contain migration histories and are hence not well-suited to study migration. Our choice to focus on individuals that have not moved since their primary school ages allows us to study non-displaced populations, while more precisely estimating local effects and mechanisms. Education is likely to be sensitive to local effects, making this a suitable choice—other work studying the effect of violence on schooling similarly focuses on non-migrants [13]. Due to the complexities of the relationship between migration, conflict and education, it is difficult to speculate as to specific directions in which the exclusion of migrants might skew our findings. In robustness checks we find smaller but still significant effects when including migrant populations; however, further work needs to be done to fully investigate the impacts of migration and displacement.

Second is the issue of survivor's bias. If the schooling of boys is disrupted to enlist in the armed forces, or if children are abducted and subsequently killed, we would underestimate the effect of conflict on education. If child soldiering is widespread among boys, the estimated null effects on surviving boys may be biased towards zero. Given the low intensity of the majority of conflict exposure in the data, this is not likely to be a major issue.

Third, while the number of years of schooling is a standard measure for education-based human capital, recent work has made a distinction between schooling and learning and highlighted region-specific gaps between the two [49]. If conflict were to impact learning rather than schooling, this would not be reflected in our estimates. New measures such as Learning-Adjusted Years of Schooling have been developed [50] and this limitation can likely be addressed in future work. Finally, there are several other limitations that are inherent to the DHS and conflict data, that are well-known and common to studies using similar data. An important one is that these are observational data, which require several assumptions for a causal interpretation of estimates. In particular, if there are remaining time-varying factors within DHS clusters that are unaccounted for, that are correlated with both conflict and education, this would limit the causal interpretation of our estimates. We discuss other data-related limitations in *Materials and methods*.

To conclude, using geo-coded conflict and survey data sets that cover large geographical areas allows us to precisely quantify the effect of local exposure to armed conflict over the entire African continent. While broad in scope, the granularity of the data allow us to draw nuanced conclusions about the effect of conflict exposure on female education. This information can be used to design more targeted interventions to mitigate the negative impacts of conflict on education. This study contributes to the literature on indirect costs of conflict, and the widespread and often overlooked toll on non-combatants. We are optimistic that a more complete understanding of the consequences of conflict can help in the design of effective approaches to deliver critical services and aid, and increase efforts to end and prevent future armed conflict.

## Materials and methods

### Educational outcomes data

Data on educational outcomes are from the Demographic and Health Surveys (DHS) (https://www.dhsprogram.com/data/). The DHS are nationally representative surveys on health and population in developing countries. Data from over 90 countries have been collected from over 350 surveys since the program started in 1984. Surveys are conducted approximately every five years, typically including between 5,000 and 30,000 households [51]. Surveys use a

two-stage sampling design, where "clusters," roughly census enumeration areas, are selected, then individual households are sampled from clusters. Information on the household and household members is first collected from any knowledgeable person age 15 or older, then eligible women (aged 15–49) and men (aged 15 to 49, 54 or 59) are interviewed individually. Our main measure of educational attainment is the years of education (in single years), collected from individual surveys. Note that the number of individual survey records for men is smaller than that for women, since in some cases, individual surveys were only conducted for women and not men. Further, a subset of surveys use "ever-married samples," where men are eligible for interview only if they have ever been married or lived in a consensual union.

Most recent surveys (and approximately 55% of all available surveys) are georeferenced, meaning that enumerators record the GPS coordinates of the center of the cluster. For respondent confidentiality, the latitude and longitude positions are randomly jittered before release, with urban cluster locations displaced by up to 2 kilometers, and rural clusters by up to 5 or 10 kilometers (for 1% of rural clusters). We consider all available georeferenced surveys from 1986 to 2022.

## Violent events data

Data on violent events are obtained from the Uppsala Conflict Data Program (UCDP) [5]. We use UCDP Georeferenced Event Dataset Global version 23.1, available at https://ucdp.uu.se/downloads/. This is an open-source collection of data on armed conflict and organized violence, collected from media reports, and is widely used in conflict research. The inclusion criteria of an event is that it must be attributed to an organized actor and result in a highest reliable estimate of at least one direct death [30]. For each recorded violent event, information on the specific location and temporal duration are available. We use data from 1989 to 2022. 31,973 violent events in the 35 African countries with DHS data were recorded, where the event location is known at least to a 25 km radius (Fig 1A). We discard the remaining 24% of events without this level of precision.

## Measuring conflict exposure and educational outcomes

To measure conflict exposure and subsequent educational outcomes, we combine the DHS and UCDP data spatially and temporally. In African countries with both DHS and UCDP data, there are a total of 118 DHS surveys with interviews of 1,938,424 individuals (men and women) from 56,623 clusters. The locations of clusters and violent events are displayed in Fig 1A.

The measure of conflict exposure we use is whether or not individuals experienced violent events within 25 km of their cluster geo-coordinates, during the ages of 6–12. Similar studies have employed smaller (10 km), the same, or larger radii (50 km) [15, 18, 19]. The appropriate size of radius depends on the outcome being studied, and is constrained by the precision of the data. For example, for mortality and immunizations, a plausible local effect might cover the distance one could travel to a healthcare facility; 25 km and 50 km have been used with the justification that those are roughly the distances a person can walk or reasonably travel to in a day [18]. For schooling outcomes, smaller radii are suitable since distances to schools tend to be smaller than distances to healthcare facilities. In a study of 8 countries in Sub-Saharan Africa, 73% of school-age children had schools within 2 km from their homes (circa 2003) [26]. Due to limitations on the precision of cluster geo-coordinates however, which are displaced by up to 10 km (*Educational outcomes data*), as well as the locations of violent events, some of which are recorded as being within 25 km of a known point (*Violent events data*), radii smaller than 25 km are not used. Using a 10 km radius, for example, would result in the size of the

measurement error potentially being larger than the local effect being studied. In *Robustness checks*, we show that results are similar using a 50 km radius.

We measure outcomes at late adolescence, age 15–18, and create separate samples for females and males. At the ages of 15–18, primary education is mostly complete. To accurately measure conflict exposure at age 6–12, we need to know the location of the child at these ages. DHS data contain locations at the time of the survey, and a question on the number of years lived in the current residence (79 out of the 118 surveys we consider asked this question). We focus on respondents who responded that they had not moved at least since age 6 (67% of female respondents aged 15–18, and 75% of males). This requires the respondents' families to have not moved, and for the respondent to still be cohabitating with older generation members. Cohabitation rates at these ages are high in Africa (around 82–91% [22]), minimizing selection biases; similar age ranges have been used in studies on educational outcomes in Africa. Further, educational attainment for women starting at the reproductive age of 15 is directly relevant to maternal, newborn, and child health [21], and is a suitable starting age to study labor market outcomes [52]. In *Robustness checks* we test the sensitivity of our results to the length of residence condition.

A full list of countries and years surveyed, along with summary statistics on the number of individuals aged 15–18, number of DHS clusters, and proportion exposed to conflict during their primary school years, is in S1 Table.

## Estimation framework

To estimate the relationship between conflict exposure and the number of years of schooling, while controlling for factors that might influence conflict and schooling, we use the following regression framework.

$$y_{ilcmt} = \beta_1 A_{lct} + \boldsymbol{\beta} \mathbf{X}_{ilct} + \gamma_{lc} + \lambda_{ct} + \tau_{cm} + \epsilon_{ilcmt} \qquad (1)$$

The outcome variable, $y_{ilcmt}$, is the education in single years for individual $i$ in DHS cluster $l$, country $c$, and is indexed to the individual's birth month $m$ and birth year $t$. $A_{lct}$ is an indicator for conflict exposure, indexed to an individual's cluster and birth year. This is a binary variable for whether or not the cluster experienced violence within 25 kilometers when the child was between the ages of 6 and 12.

We include an array of individual and time-varying cluster-level characteristics, $\mathbf{X}_{ilct}$. Individual-level controls include household size, age and sex of the head of household, whether the mother is living in the household, and household wealth index. Time-varying cluster-level controls include mean nightlight intensity (https://figshare.com/articles/dataset/Harmonization_of_DMSP_and_VIIRS_nighttime_light_data_from_1992-2018_at_the_global_scale/9828827/8) [53], rainfall and temperature (https://developers.google.com/earth-engine/datasets/catalog/IDAHO_EPSCOR_TERRACLIMATE) when the child was at age 6. $\gamma_{lc}$ are location fixed effects at the cluster level, and $\lambda_{ct}$ are time fixed effects at a country-birth year level. $\tau_{cm}$ are additional country-birth month fixed effects that flexibly capture differences in educational outcomes related to birth month. $\epsilon_{ilcmt}$ is the error term. Estimation is done using ordinary least squares, and all standard errors are clustered at the DHS cluster level, which adjusts for correlations between individuals in the same DHS cluster. Following DHS recommendations, we weight observations by the product of survey-specific weights and the inverse of the fraction of the relevant population interviewed, so that estimates are representative of all the countries in our sample [54, 55]. Age-specific yearly population [56] is from https://population.un.org/wpp/Download/Files/\1_Indicators%20(Standard)/CSV_FILES/WPP2022_PopulationByAge5GroupSex_Medium.zip.

The main identifying assumption is that there are no unmeasured confounders that affect both the occurrence of violence when the child is age 6–12, and the years of schooling by age 15–18. Cluster fixed effects control for unobserved cluster-level characteristics that may affect both the occurrence of conflict and the years of schooling, so that cross-sectional comparisons are avoided. Time fixed effects control for common effects over time (birth year), for example economic prosperity in certain years resulting in fewer conflicts and larger investments in educational infrastructure. We allow these time fixed effects to be country-specific. Other controls include nightlight intensity (which is correlated with economic activity), temperature and rainfall, which could plausibly affect both the occurrence of conflict and school enrollment or attendance. Consistent with the inclusion of these fixed effects and controls, we use causal language to describe the effects of conflict [15], but acknowledge the limitations of using observational data for this purpose.

Individual-level controls and country-birth month fixed effects $\tau_{cm}$ may affect the outcome variable and are included to improve the precision of estimates. The latter account for month-of-birth effects on educational attainment, for example that children born later in the year have poorer educational outcomes [57].

We run separate regressions for females and males. Males and females are treated separately for several reasons. First, the sampling processes for women and men are slightly different (see *Educational outcomes data*). Second, confounding effects could be in opposite directions for females and males. For example, some locations may be more likely to have both conflict and higher male years of schooling, but lower female years of schooling (or vice versa). A single location fixed effect for both sexes will not do an adequate job of controlling for these effects. A formal test for whether the data should be combined (e.g., a Chow test), rejects the null hypothesis that a combined model is better.

## Robustness checks

We check the robustness of our results to sample construction conditions and modeling assumptions. In columns (1)-(2) of S2 Table, we remove the condition requiring respondents to have lived in their current residence at least since age 6. The resulting sample includes (A) individuals from surveys that did not ask the question of length of residence, (B) individuals who lived in their current residence at least since age 6 (our main sample), and (C) individuals who had moved since age 6. Columns (3) and (4) display the main results in Fig 2A (sample B), and columns (5) and (6) are for the subset that recently moved (sample C).

In columns (1)-(2), we find a smaller but still significant negative effect for females, and a positive but insignificant effect for males. In column (5), the estimated effect is -0.11 and not significant. This reflects a potential negative impact for migrants that is attenuated by measurement error in conflict exposure. Our ability to accurately measure conflict exposure for those who moved is limited to respondents whose previous residences are close enough to their current residence.

S3 Table tests the sensitivity of the main results to modeling assumptions. Column (1) uses a distance from conflict threshold of 50 km instead of 25 km. Column (2) removes household-level covariates that are measured at the time of the survey. These may have changed since the treatment or be endogenous to the treatment. Column (3) removes sample weights, which may be sensitive to the population estimates used [58]. Column (4) uses a sample that is conditional on not having conflict in the survey year and the year before the survey, to exclude the possibility that effects are influenced by more recent or current conflict. In all four cases, the estimated coefficient for the effect of conflict exposure to female years of schooling remains negative and significant.

Finally, S4 Table shows results for a binary outcome variable for whether or not the female respondent has completed at least a primary education, an outcome that is used in some studies of the effect of conflict on education (e.g., [10, 36]). We find a similar negative effect.

## Distance from conflict

To investigate the effect of distance from conflict, we use the following regression equation for the female sample.

$$y_{ilcmt} = \sum_{d=1}^{4} \beta_d A_{lct}^d + \boldsymbol{\beta} \mathbf{X}_{ilct} + \gamma_{lc} + \lambda_{ct} + \tau_{cm} + \epsilon_{ilcmt} \tag{2}$$

The treatment indicators are $A_{lct}^d$, for distance thresholds $d$ from 1 to 4, representing conflict occurring 0–25 km, 25–50 km, 50–75 km and 75–100 km from the cluster geo-coordinates, when the child was between the ages of 6 and 12.

## Conflict at different ages

To analyze the impact of conflict exposure at individual ages, we use the following specification for the female sample.

$$y_{ilcmt} = \sum_{a=6}^{12} \beta_a A_{lct}^a + \boldsymbol{\beta} \mathbf{X}_{ilct} + \gamma_{lc} + \lambda_{ct} + \tau_{cm} + \epsilon_{ilcmt} \tag{3}$$

The treatment indicators are $A_{lct}^a$, for ages $a$ from 6 to 12, representing conflict occurring within 25 km from the cluster geo-coordinates, in the calendar year when the child turned age $a$.

In the additional analysis presented in S1 Fig, we consider binary outcome variables of whether or not the respondent had no education, and whether or not the respondent had an incomplete primary education. These are based on a categorical educational attainment variable in the DHS, rather than the numerical education in single years used in the main analysis. Categories available are: none, incomplete primary, complete primary, incomplete secondary, complete secondary, higher education. The specification is as follows, and we perform the estimation using logistic regression for the female sample.

$$g(\mathbb{E}[y_{ilcmt}]) = \sum_{a=6}^{12} \beta_a A_{lct}^a + \boldsymbol{\beta} \mathbf{X}_{ilct} + \gamma_{lc} + \lambda_{ct} + \tau_{cm} \tag{4}$$

where $y_{ilcmt}$ is an indicator that is 1 if the individual $i$ had completed no education, and 0 otherwise, and similarly for having incomplete primary education. $g()$ represents the logit transformation.

## Chronicity and intensity of conflict

To characterize the chronicity and intensity of conflict exposure, we consider the number of events the individual was exposed to, the associated total number of casualties, and the number of years of exposure to violence during the ages of 6 to 12. To estimate the effects of different types of violence, we consider exposures of the lowest intensity of up to 2 casualties, the bottom quintile of the number of casualties experienced. For conflict exposure at each age from 6–12, we define recurring exposures to be when the individual was also exposed to conflict events in the preceding year, and new exposures, where no events were recorded in the previous year. We then use specifications similar to Eqs 2 and 3 for different types of conflict experiences, for the female sample.

## Location heterogeneity

To allow for the possibility that the effect on female years of schooling might differ for different sub-populations, we estimate Eq 1 using subsets of DHS clusters with different characteristics: rural vs. urban, relative wealth quartile, "baseline" levels of educational attainment, gender gap in educational attainment, and geographical region.

The rural-urban classification is based on the DHS. For wealth, we divide the sample into quartiles based on relative wealth levels. The DHS wealth index is a categorical classification into five levels, based on asset ownership. We calculate the mean wealth index among all households in each cluster, and divide these into quartiles among the regression sample. For educational attainment, we similarly divide the sample into quartiles based on educational attainment of older females. For each cluster, we consider females aged 40–50—these are the oldest women interviewed in the surveys, approximately corresponding to the generation of mothers to the adolescents whose educational outcomes we consider. We calculate the mean years of schooling among these women for each cluster, and then divide the regression sample into quartiles. The number of DHS clusters in each of the wealth and education quartiles is in S2 Fig.

Since these variables are measured at the time of the survey and not at the time of conflict exposure, a potential issue is that they are affected by conflict. This might affect the wealth variable in particular: recent local conflict may have a detrimental effect on household wealth. We perform a robustness check using nightlight intensity at age 6 instead of the DHS wealth index (at the time of the survey) to measure wealth. Previous work has shown that nightlight intensity can be used as a proxy for economic activity [59, 60]. We first use nightlight intensity at the year of the survey to replicate the results in Fig 4, and repeat the exercise using nightlight intensity at *age 6* (S3 Fig).

For gender gap in educational attainment, we calculate the mean years of schooling among all males and females sampled in each administrative level 2 region. The difference (males—females) is the gender gap; these are plotted in S4 Fig. The median gender gap among administrative level 2 regions is 1.2 years. We then split DHS clusters into whether they are located in a region with gender gap above or below the median.

Finally, for geographical regions, we use regions defined in the GBD study. These regions are designed to be geographically close and similar in socio-demographics and epidemiology, and have been used in related work [21]. Africa is divided into five regions: North Africa, Central, Eastern, Southern and Western Sub-Saharan Africa (SSA). We focus on Central, Eastern and Western SSA, the three regions with both higher levels of conflict and larger number of individuals surveyed.

## Mechanisms

To investigate mechanisms by which conflict exposure affects female schooling outcomes, we consider additional outcome variables that are derived from questions in the DHS: whether or not the respondent has ever been in a union (married), from the individual survey, and whether or not the respondent's mother and father are alive, from the household survey. In S5 Table we model these binary outcome variables using a logistic regression, with the same variables as in the main specification in Eq 1.

$$g(\mathbb{E}[y_{ilcmt}]) = \beta_1 A_{lct} + \boldsymbol{\beta} \mathbf{X}_{ilct} + \gamma_{lc} + \lambda_{ct} + \tau_{cm} \tag{5}$$

Since these outcome variables are likely to be sensitive to the length of residence condition, i.e., that respondents lived in their current residence since age 6 (e.g., those that are married

are less likely to be cohabitating with an older generation member), we additionally perform a sensitivity check that removes this condition (S6 Table).

## Mortality

To calculate excess deaths, we first estimate annual child mortality rates in Africa from 2000–2019, under the conflict-free counterfactual. We use a rate of conflict exposure among mothers of 21%, the same as in our sample, and a reduction of years of schooling for this group of 0.38 years, as estimated in our main results. For the relationship between years of schooling and child (under-5) mortality, we use estimates from a recent global systematic review and meta-analysis by Balaj et al., which found that each additional year of schooling for mothers was associated with a 3.04% reduction in under-5 mortality, on average [35]. We use observed child mortality rates from the 2019 Global Burden of Disease (GBD) study [29] for all countries in Africa in 2000–2019.

Next, to translate the counterfactual mortality rates to total number of counterfactual under-5 deaths, we multiply the rates by the number of children (from GBD estimates). The number of excess deaths is then the difference between the number of observed under-5 deaths, and the (lower) number of counterfactual deaths.

Low and high estimates are obtained using 95% C.I.s of the Balaj et al. estimates and our estimates, and lower and upper estimates of GBD child mortality rates. The number of direct under-5 casualties due to conflict and terrorism are GBD estimates, together with lower and upper estimates.

The above back-of-the-envelope estimates require several assumptions. The first is that there is no selection into conflict or motherhood. This means that mothers and non-mothers are equally likely to be exposed to conflict—if those that experienced conflict are less likely to become mothers, then excess deaths would be overestimated. The effect of conflict on years of schooling is assumed to be the same for mothers and non-mothers, and observed childbirth and child mortality rates are the same for conflict-exposed and non-exposed mothers. If conflict-exposed mothers have higher child mortality, excess deaths would be underestimated.

The second assumption is that the number of years of schooling lost at age 15–18 persists to motherhood. We show in S7 Table that conflict exposure does not significantly change the likelihood of school attendance at age 15–18; this suggests that delays in schooling are accounted for by age 15–18, limiting the concern that conflict-related losses in schooling are reversed later in life. The third assumption is that our estimated effects apply to older birth cohorts. Our analysis was based on the 1983–2007 birth cohorts, while mothers in 2000–2019 may be from slightly older birth cohorts; our estimates are valid if the experiences of older and younger birth cohorts is similar. We show in S8 Table that both conflict incidence and effect sizes are similar between earlier (1983–1996) and later (1997–2007) birth cohorts, suggesting that estimates are relatively stable over time. The final assumption is that estimated effects apply to other countries in Africa that data were not available for.

## Other limitations

There are several other limitations associated with the data that we use. For the DHS data, since cluster locations are randomly displaced, conflict exposure may not be accurately measured and estimates would be biased towards zero. DHS data may miss areas affected by conflict and violence [8]; these would bias estimates towards zero. We measure conflict exposure during primary school ages and not during the time of the survey, limiting this concern.

The UCDP data are collected from media reports, and are biased towards populous regions, economic centers, areas with better infrastructure, and more salient events. Imprecision in

geo-coordinates might introduce measurement error, biasing estimates towards zero. A type of bias that is relevant to our analysis is that the threshold for reporting might differ in accessible versus inaccessible areas, where in inaccessible areas, occurrences of violence have to be more serious in order to be considered "newsworthy" [61]. In our analysis on rural versus urban areas, if the events in rural areas are more serious because of reporting biases, then the effect of violence may be overestimated. We do not find differing impacts for rural and urban areas, but if this hypothesis were true, the effects in urban areas might be more severe than in rural areas. Unfortunately it is not possible to directly test this hypothesis, as it would require information on violent events that are not recorded. Nevertheless, UCDP compares favorably to other data sets such as the Armed Conflict Location and Event Data Project (ACLED) in terms of spatial and temporal coverage, as well as accuracy of the data [62]. Future work might address some of these challenges. For example, the limitations of media-based conflict data are now widely known [63] and there are best practices available to address them [64].

## Ethics statement

The study used secondary data from the Demographic and Health Surveys (DHS). Procedures and questionnaires for standard DHS surveys have been reviewed and approved by the ICF Institutional Review Board. Informed consent was sought from each survey participant. Administrative permissions for accessing the DHS data for this study were obtained from the DHS program, and use of the data is consistent with its terms of use (https://dhsprogram.com/Data/terms-of-use.cfm). Since this study is based on de-identified and anonymous data available for secondary analyses, no further ethical approval was necessary.

## Supporting information

**S1 Table. Countries and survey years of the Demographic Health Surveys included in the main results.** Summary statistics on the respondents (male and female, age 15-18) in each survey year, including the number of clusters they reside in, and the proportion of these respondents that were exposed to any conflict during the ages of 6-12.
(PDF)

**S2 Table. Robustness checks for the overall effect of conflict exposure on years of schooling for females and males using different regression samples.** Columns (1)-(2) do not include any constraints on the length of stay in current residence. The sample thus includes both non-migrants and migrants. Columns (3)-(4) display the main results in Fig 2A. These incorporate the residence condition (not moving since age 6). Columns (5)-(6) includes only individuals who had moved since age 6. All models include cluster, country-birth year, and country-birth month fixed effects. Standard errors are clustered at a DHS cluster level. *p<0.1; **p<0.05; ***p<0.01.
(PDF)

**S3 Table. Sensitivity of the overall effect of conflict exposure on female years of schooling to modeling assumptions.** Column (1) uses a 50 km radius for conflict exposure, instead of 25 km. Column (2) removes household characteristics. Since these characteristics, taken from survey data, are measured at the time of the survey, they may have changed since the treatment (school-age conflict exposure) or may have been affected by the treatment. Column (3) removes sample weights. Sample weights are necessary to ensure that estimates are representative of all the countries in the sample, but may be sensitive to the population estimates used. Column (4) only includes observations that did not have conflict in the survey year and the year before the survey, to exclude the possibility that effects are due to more recent conflict. All

models include cluster, country-birth year, and country-birth month fixed effects and use the same female sample as the main results (S2 Table, column (3)). Standard errors are clustered at a DHS cluster level. *p<0.1; **p<0.05; ***p<0.01.
(PDF)

**S4 Table. Alternative outcome variable for educational attainment.** Regression uses a binary variable for whether or not the individual has completed at least a primary education, as is sometimes used in the literature, instead of Years of schooling. Coefficient estimates are from logistic regression on the female sample in the main results (S2 Table, column (3)). Standard errors are clustered at a DHS cluster level. *p<0.1; **p<0.05; ***p<0.01.
(PDF)

**S5 Table. Mechanisms for the effect of conflict exposure on female years of schooling.** Regressions use different outcome variables for different potential mechanisms: column (1) uses a binary variable taking the value 1 if the individual has never been married. Column (2) uses a binary variable taking the value 1 if the individual's mother is still alive, and column (3) for the father. Columns (2) and (3) remove the controls for female head of household and whether or not the mother is in the household, since these are highly correlated with the outcome variables. Coefficient estimates are from logistic regression on the female sample in the main results (S2 Table, column (3)). Standard errors are clustered at a DHS cluster level. *p<0.1; **p<0.05; ***p<0.01.
(PDF)

**S6 Table. Sensitivity of the results in S5 Table to the length of residence condition.** The outcome variables in S5 Table potentially affect whether or not an individual is still cohabitating with a member of the older generation. As explained in *Materials and Methods*, the length of residence condition, used in the main sample (and in S5 Table), requires that individuals still be cohabitating with the older generation. Regressions in this table remove the length of residence condition (corresponding to overall effects in columns (1)-(2) of S2 Table), so individuals who moved since age 6 are included in the sample. The effect of conflict exposure remains insignificant, and do not change the conclusion that there is no evidence to suggest that conflict exposure increases the likelihood of child marriages or parental deaths. Coefficient estimates are from logistic regression. Standard errors are clustered at a DHS cluster level. *p<0.1; **p<0.05; ***p<0.01.
(PDF)

**S7 Table. Effect of conflict exposure on school attendance at age 15-18.** The outcome variable in the regression is based on responses to a question on current school attendance in the household portion of the DHS. The regression outcome variable is coded 1 if the respondent was currently attending or attended school at some time in the current school year. If it were the case that schooling is delayed due to conflict, so losses in schooling might be reversed by adulthood, we would expect positive effects on school attendance at age 15-18. Here we do not find significant effects. Coefficient estimates are from logistic regression on the female sample in the main results (S2 Table, column (3)). Standard errors are clustered at a DHS cluster level. *p<0.1; **p<0.05; ***p<0.01.
(PDF)

**S8 Table. Effect of conflict exposure on female years of schooling for different birth cohorts.** Column (1) contains the subset of the female sample, used in the main results, that was born in 1983-1996. Column (2) contains the subset that was born in 1997-2007. The rates of conflict exposure in the earlier and later birth cohorts are similar at 21.2% and 21.6%. All

models include cluster, country-birth year, and country-birth month fixed effects. Standard errors are clustered at a DHS cluster level. *p<0.1; **p<0.05; ***p<0.01.
(PDF)

**S1 Fig. Effect of conflict exposure at individual ages on female years of schooling.** Conflict in each year is defined as the occurrence of a violent event within 25 km in each year that the respondent turned 6 to 12. Exponentiated regression coefficients, plotted on the y axis, indicate the multiplicative increase in the odds of having no education (top), and incomplete primary education (bottom). Estimation is by logistic regression and bars are 95% confidence intervals.
(TIF)

**S2 Fig. Number of DHS clusters in each wealth and education quartile.** Wealth quartiles are based on cluster mean relative wealth index. Education quartiles are based on cluster mean years of schooling for females aged 40-50.
(TIF)

**S3 Fig. Robustness check for heterogeneous effects for wealth and education.** Top: wealth quartiles are based on nightlight intensity at the time of the survey, rather than DHS wealth index at the time of the survey in Fig 4. Bottom: wealth quartiles are based on nightlight intensity at age 6. Measuring wealth at age 6 instead of at the time of the survey addresses the concern than wealth may be affected by conflict occurrence and is thus endogenous. Here we see that the results are qualitatively unchanged. Bars are 95% confidence intervals.
(TIF)

**S4 Fig. Gender gap between males and females.** Gender gap is the difference between mean years of schooling among all males and females surveyed in each administrative level 2 region. Regions with no data are colored in white. Administrative boundaries are from [20].
(TIF)

## Acknowledgments

I thank participants of the 19th Annual Households in Conflict Workshop, 21st Midwest International Economic Development Conference and the South-East Exchange of Development Studies (SEEDS) 4th Annual Conference for helpful comments and suggestions.

## Author Contributions

**Conceptualization:** Xiao Hui Tai.

**Data curation:** Xiao Hui Tai.

**Formal analysis:** Xiao Hui Tai.

**Methodology:** Xiao Hui Tai.

**Software:** Xiao Hui Tai.

**Writing – original draft:** Xiao Hui Tai.

**Writing – review & editing:** Xiao Hui Tai.

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
