## [Decision Letter · Decision Letter 0]

6 Sep 2024

PONE-D-24-25168

Nearby Armed Conflict Affects Girls' Education in Africa

PLOS ONE

Dear Dr. Tai,

Thank you for submitting your manuscript to PLOS ONE. After careful consideration, we feel that it has merit but does not fully meet PLOS ONE’s publication criteria as it currently stands. Therefore, we invite you to submit a revised version of the manuscript that addresses the points raised during the review process.

We look forward to receiving your revised manuscript.

Kind regards,

Rabie Adel El Arab

Academic Editor

PLOS ONE

Journal Requirements:

2. We note that Figure 1 in your submission contain map/satellite images which may be copyrighted. All PLOS content is published under the Creative Commons Attribution License (CC BY 4.0), which means that the manuscript, images, and Supporting Information files will be freely available online, and any third party is permitted to access, download, copy, distribute, and use these materials in any way, even commercially, with proper attribution. For these reasons, we cannot publish previously copyrighted maps or satellite images created using proprietary data, such as Google software (Google Maps, Street View, and Earth). For more information, see our copyright guidelines: http://journals.plos.org/plosone/s/licenses-and-copyright.

a. You may seek permission from the original copyright holder of Figure(s) [#] to publish the content specifically under the CC BY 4.0 license.  

5. We note that there is identifying data in the Supporting Information file <ConflictEducation_SM>. Due to the inclusion of these potentially identifying data, we have removed this file from your file inventory. Prior to sharing human research participant data, authors should consult with an ethics committee to ensure data are shared in accordance with participant consent and all applicable local laws. Data sharing should never compromise participant privacy. It is therefore not appropriate to publicly share personally identifiable data on human research participants. The following are examples of data that should not be shared: -Name, initials, physical address -Ages more specific than whole numbers -Internet protocol (IP) address -Specific dates (birth dates, death dates, examination dates, etc.) -Contact information such as phone number or email address -Location data -ID numbers that seem specific (long numbers, include initials, titled “Hospital ID”) rather than random (small numbers in numerical order) Data that are not directly identifying may also be inappropriate to share, as in combination they can become identifying. For example, data collected from a small group of participants, vulnerable populations, or private groups should not be shared if they involve indirect identifiers (such as sex, ethnicity, location, etc.) that may risk the identification of study participants. Additional guidance on preparing raw data for publication can be found in our Data Policy (https://journals.plos.org/plosone/s/data-availability#loc-human-research-participant-data-and-other-sensitive-data) and in the following article: http://www.bmj.com/content/340/bmj.c181.long. Please remove or anonymize all personal information (<specific identifying information in file to be removed>), ensure that the data shared are in accordance with participant consent, and re-upload a fully anonymized data set. Please note that spreadsheet columns with personal information must be removed and not hidden as all hidden columns will appear in the published file.

Additional Editor Comments:

Dear Author,

Thank you for submitting your manuscript. I added below few additional comments you might consider / clarify

1. Interpretation of Results:

Clarification of Habituation to Violence: The interpretation of recurring conflict leading to a "habituation to violence" and its positive impact on education requires further exploration. It would be beneficial to discuss alternative explanations for this finding, such as changes in community resilience, school infrastructure, or other socio-political dynamics. Consider examining these factors more closely to ensure that the interpretation is well-grounded.

Effect of Single Low-Casualty Events: The significant negative impact of a single conflict event with minimal casualties on girls' education could be influenced by factors other than the immediate violence. Please consider whether measurement errors, socio-economic factors, or other psychological impacts could be contributing to this result and address these possibilities in the discussion.

2. Estimation and Data Sensitivity:

Distance Measure for Conflict Exposure: The use of a fixed 25 km radius to define conflict exposure, while practical, may oversimplify the complex ways in which conflict impacts education across different terrains and population densities. We recommend providing a more detailed justification for this choice or exploring how varying the distance might affect your results.

Potential Overestimation of Child Mortality Impact: The back-of-the-envelope calculation estimating excess under-5 deaths due to reduced female education may be overestimated due to broad assumptions. Please consider refining this analysis by accounting for potential variations in healthcare access, cultural practices, and local interventions that might mediate the relationship between maternal education and child mortality.

3. Missing Considerations:

Inclusion of Migrant Populations: The exclusion of migrants and displaced individuals is a significant limitation that may lead to underestimation of the conflict's overall impact on education. We suggest either including this population in your analysis or providing a more comprehensive discussion on how their exclusion might skew your findings.

Broader Socioeconomic Variables: While wealth and education quartiles are used to assess location-specific heterogeneity, other important socioeconomic variables, such as parental education, household income stability, and access to educational resources, should also be considered. Expanding the range of socioeconomic factors analyzed could provide a more nuanced understanding of how conflict affects education.

4. Potential Data Biases:

Media-Based Data Collection: The reliance on media reports for conflict event data may introduce biases, particularly in underreported or less accessible regions. Please discuss how these potential biases might impact your findings and consider conducting additional robustness checks to account for this.

5. Generalization and Policy Implications:

Overgeneralization Across Diverse Regions: Given the vast cultural, political, and economic diversity within Africa, generalizing findings across 28 countries may overlook country-specific factors. We recommend either providing a more localized analysis or discussing how your results might vary across different regions.

Policy Recommendations: The manuscript currently lacks concrete policy recommendations. Including actionable suggestions for policymakers or interventions to mitigate the impact of conflict on education would enhance the practical relevance of your study.

6. Mechanisms of Impact:

Detailed Exploration of Mechanisms: The discussion of potential mechanisms through which conflict impacts girls’ education is somewhat speculative. We recommend conducting additional analyses to empirically test these mechanisms or, at the very least, expanding the discussion to consider them in greater detail.

Best wishes

Reviewers' comments:

Reviewer's Responses to Questions

**Comments to the Author**

1. Is the manuscript technically sound, and do the data support the conclusions?

Reviewer #1: Yes

Reviewer #2: Partly

Reviewer #3: Yes

2. Has the statistical analysis been performed appropriately and rigorously? 

Reviewer #1: Yes

Reviewer #2: Yes

Reviewer #3: Yes

3. Have the authors made all data underlying the findings in their manuscript fully available?

Reviewer #1: Yes

Reviewer #2: Yes

Reviewer #3: Yes

4. Is the manuscript presented in an intelligible fashion and written in standard English?

Reviewer #1: Yes

Reviewer #2: Yes

Reviewer #3: Yes

5. Review Comments to the Author

Reviewer #1: Overall, this Research Article is a compelling treatment of the complex relationship between the impact of conflict on female

educational attainment in 28 African countries. It is rich with data, data analysis -- plus justification of data selected and systematically applied. The lone author clearly states the purpose of the research, explains his methodology with insightful details -- and remains on course to results and discussion with clear cut regression analysis. The research presented is technically sound with logical justification -- and reveals and confirms a measurable loss of educational attainment (0.4 years) for females in selected African countries having experienced conflict. Also, the Research Article is a timely and reasonable body od knowledge for developing policy to mitigate the impact of conflict on educational attainment in struggling countries.

Reviewer #2: Thank you for giving me the opportunity to review this interesting paper. The study aims to estimate the impact of local conflict events on the educational attainment of girls in Africa. The writing and presentation are commendable. However, I have one major concern and one minor comment. With some improvements, I believe this paper could be of great interest to the journal's readership.

Major

As the author is aware, the paper estimates correlation, which does not necessarily imply causation. While several robustness checks have been performed, the study fails to establish of a clear counterfactual. I understand that some may claim causality using OLS with strong assumptions, but it would be beneficial for the author to explore exogenous variation in the treatment. For example, a Regression Discontinuity Design (RDD) using distance to the conflict event as the running variable, with 25 km as the cut-off, could be considered (this is just an example). This is a significant drawback of the study. If the author decides to proceed with the current estimation strategy, they should clearly state the strategy, its limitations, and the strong assumptions made to claim causality in the abstract, introduction, and conclusion sections.

Minor

Typically, readers expect to see the materials and methods section before the results and discussion. The author may want to restructure the paper so that it follows this sequence, 1. Introduction, 2. Materials and Methods.

Reviewer #3: This is a very interesting paper looking at the impact of proximity of armed conflict on girls’ educational attainment in terms of years of education. The paper uses two relatively large data sets on 28 African countries: the survey data comes from DHS (Demographic and Health Surveys) and the conflict from Uppsala Conflict Data Programme (UCDP). A competent econometric analysis with many robustness checks has been conducted. The paper also contains a succinct literature review. Overall, this is an important contribution to the literature on the impact of conflict on education with particular focus on female primary education.

The following few paragraphs raises some issues that need clarification.

One of the interesting and intriguing finding of the paper is the habituation to violence. This is summarised by the following quote from page 3: “On the other hand, recurring violence, defined as events occurring in consecutive years, has a positive effect at certain ages of exposure, suggesting a habituation to conflict.” It is difficult to understand how consecutive years of conflict positively affect years of education; as far as we are considering marginal effects of conflict (i.e., controlling for other factors that affect conflict), shouldn’t it be necessarily negative (or at best zero)?

p. 14: “We measure outcomes at late adolescence, age 15-18, and create separate samples for females and males.” Why estimate separate regressions for females and males? Isn’t the better option to run on pooled data with female/male dummy? When running separate regressions, the degrees of freedom in the two samples may differ influencing the estimated results; a pooled regression with dummy variables can avoid that. Alternatively, if there is reason to suspect that female and male years of education are systematically different, it is better to run a formal test for structural break (e.g., Chow test) to justify the estimation of separate regressions.

p. 15: “We include an array of individual and time-varying cluster-level characteristics, Xilcmt, …” How can this vector of variables be indexed to month and year of the individual’s birth (since we have subscript mt) unless DHS surveys specifically ask questions for each individual corresponding to condition existing at their birth? X includes demographic variables like household size, age and sex of household head, etc. Since the Xs are indexed mt, this implies all these variables refer to the value of the Xs at a month and year the individual is born, i.e., household size, age/sex of household head, etc., when the girl/boy was born. This is unlikely. This needs clarification.

p. 22: “The fixed effects and other covariates in our model control for location-specific and household-specific factors, time trends, and time-varying characteristics such as nightlights and temperature …” This statement gives the impression that the regressions control for household fixed effects. Isn't DHS data repeated cross-section? If so, you cannot control for household-fixed effects. Clarify or change the statement.

6. PLOS authors have the option to publish the peer review history of their article (what does this mean?). If published, this will include your full peer review and any attached files.

Reviewer #1: No

Reviewer #2: No

Reviewer #3: No

---

## [Decision Letter · Decision Letter 1]

6 Nov 2024

Nearby armed conflict affects girls' education in Africa

PONE-D-24-25168R1

Dear Dr. Tai,

We’re pleased to inform you that your manuscript has been judged scientifically suitable for publication and will be formally accepted for publication once it meets all outstanding technical requirements.

Kind regards,

Marwa Ramadan

Academic Editor

PLOS ONE

Additional Editor Comments (optional):

Reviewers' comments:

Reviewer's Responses to Questions

**Comments to the Author**

1. If the authors have adequately addressed your comments raised in a previous round of review and you feel that this manuscript is now acceptable for publication, you may indicate that here to bypass the “Comments to the Author” section, enter your conflict of interest statement in the “Confidential to Editor” section, and submit your "Accept" recommendation.

Reviewer #1: All comments have been addressed

Reviewer #2: All comments have been addressed

2. Is the manuscript technically sound, and do the data support the conclusions?

Reviewer #1: Yes

Reviewer #2: Yes

3. Has the statistical analysis been performed appropriately and rigorously? 

Reviewer #1: Yes

Reviewer #2: Yes

4. Have the authors made all data underlying the findings in their manuscript fully available?

Reviewer #1: Yes

Reviewer #2: Yes

5. Is the manuscript presented in an intelligible fashion and written in standard English?

Reviewer #1: Yes

Reviewer #2: Yes

6. Review Comments to the Author

Reviewer #1: Kudos for your timely and thorough answers to the concerns of the reviewers. My encouragement remains unchanged. Your replies were detailed and equally candid from my perspective. I remain persuaded that your work will be a genuine eye-opener, proving fodder to the often rambling rhetoric about the impact of violence. It would be interesting to overlay your methodology to the impact of violence on life expectancy from an indefinite disease outbreak eg.Mpox in the DRC. Also, I

agree withe the Title adjustment. Moreover, your "recurring factor" continues to ring in my mind as the logo of your work.

Solid work with an unique one-person (compliment) analysis -- and my honor to have had a reviewer's role.

Reviewer #2: (No Response)

7. PLOS authors have the option to publish the peer review history of their article (what does this mean?). If published, this will include your full peer review and any attached files.

Reviewer #1: No

Reviewer #2: No

---

## [Editor Report · Acceptance letter]

22 Nov 2024

PONE-D-24-25168R1 

PLOS ONE

Dear Dr. Tai, 

I'm pleased to inform you that your manuscript has been deemed suitable for publication in PLOS ONE. Congratulations! Your manuscript is now being handed over to our production team.

Kind regards, 

on behalf of

Dr. Marwa Ramadan 

Academic Editor

PLOS ONE